# Assessment of the Influence of Survey Design and Processing Choices on the Accuracy of Tree Diameter at Breast Height (DBH) Measurements Using UAV-Based Photogrammetry

**Bruno Miguez Moreira** [1,*] , **Gabriel Goyanes** [1,2] , **Pedro Pina** [1] , **Oleg Vassilev** [3] **and Sandra Heleno** [1]

1 Centre of Natural Resources and Environment (CERENA), IST, University of Lisbon, 1049-001 Lisbon, Portugal; gabriel.goyanes@tecnico.ulisboa.pt (G.G.); ppina@tecnico.ulisboa.pt (P.P.); sandra.heleno@tecnico.ulisboa.pt (S.H.)
2 Centre of Geographical Studies (CEG), IGOT, University of Lisbon, 1649-004 Lisbon, Portugal
3 Bulgarian Antarctic Institute, 1504 Sofia, 15 Tsar Osvoboditel Boulevard, Bulgaria; ovassilev@gmail.com
* Correspondence: bruno.miguez@tecnico.ulisboa.pt

**Abstract:** This work provides a systematic evaluation of how survey design and computer processing choices (such as the software used or the workflow/parameters chosen) influence unmanned aerial vehicle (UAV)-based photogrammetry retrieval of tree diameter at breast height (DBH), an important 3D structural parameter in forest inventory and biomass estimation. The study areas were an agricultural field located in the province of Málaga, Spain, where a small group of olive trees was chosen for the UAV surveys, and an open woodland area in the outskirts of Sofia, the capital of Bulgaria, where a 10 ha area grove, composed mainly of birch trees, was overflown. A DJI Phantom 4 Pro quadcopter UAV was used for the image acquisition. We applied structure from motion (SfM) to generate 3D point clouds of individual trees, using Agisoft and Pix4D software packages. The estimation of DBH in the point clouds was made using a RANSAC-based circle fitting tool from the TreeLS R package. All trees modeled had their DBH tape-measured on the ground for accuracy assessment. In the first study site, we executed many diversely designed flights, to identify which parameters (flying altitude, camera tilt, and processing method) gave us the most accurate DBH estimations; then, the resulting best settings configuration was used to assess the replicability of the method in the forested area in Bulgaria. The best configuration tested (flight altitudes of about 25 m above tree canopies, camera tilt 60°, forward and side overlaps of 90%, Agisoft ultrahigh processing) resulted in root mean square errors (RMSEs; %) of below 5% of the tree diameters in the first site and below 12.5% in the forested area. We demonstrate that, when carefully designed methodologies are used, SfM can measure the DBH of single trees with very good accuracy, and to our knowledge, the results presented here are the best achieved so far using (above-canopy) UAV-based photogrammetry.

**Keywords:** unmanned aerial vehicle (UAV); photogrammetry; structure from motion; 3D point cloud; diameter at breast height (DBH); forest inventory; remote sensing

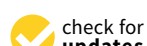



## 1. Introduction

The basis for sustainable forest management is the systematic collection of data within a given area, called a forest inventory, and its most crucial information is biomass estimation [1]. On a large spatial scale, being able to accurately map the extent and condition of forest biomass is important for climate change modeling, greenhouse gas inventories, and terrestrial carbon estimations. On a smaller spatial scale, accurate tree mass estimations are needed for commercial purposes, such as the wood industry where the stem mass is procured, or in biofuel production, where all biomass available above the ground can be used (including the tree trunk, stump, branches, foliage, and understory). Modeling the 3D structure of a forest also provides information about ecosystem dynamics, including regeneration, growth, and understory development [2–4].

Manually assessing each tree in the field is a prohibitively time-consuming task, and the motivation for enhancing and modernizing forest inventory measurements is growing, in view of the uncertainty related to risks posed to forests by anthropogenic pressure or climate change [5]. Remote sensing methods are increasingly being used by forest practitioners and scientists to retrieve critical 3D parameters of the forest (such as number of trees in a given area, tree height, trunk diameter, and single tree volume), balancing the requirements for accuracy and efficiency in biomass estimation.

Light-detection and ranging (LiDAR) is a method that uses pulses of laser energy, capable of penetrating vegetation, to provide very accurate surface and terrain 3D representations. For the last 15 years, airborne LiDAR (also named laser scanning) has been the standard tool in 3D forest mapping. However, the altitude of the manned aircraft produces relatively low-density point clouds (typically below 5 pts/m$^2$) that may not allow high-precision biomass estimations; besides, when the study area is relatively small or when repeat acquisitions are needed to provide information on forest temporal changes, perhaps one of the most critical forest management needs, its cost is usually prohibitive [5].

Small unmanned aerial vehicles (UAV), or drones, much easier to fly on-demand, can lower LiDAR survey costs and provide denser point clouds when compared to classic airborne laser scanning. UAV-borne LiDAR has recently been introduced in forest inventory and biomass estimation, with very promising results [6–8]. Additionally, terrestrial laser scanning (TLS), which differs from airborne laser scanning by the ground-based platform, has been widely used in the past decades to perform plot-level forest inventory at very high precision; for example, it has been used to estimate attributes of trees such as the diameter at breast height (DBH) or tree height [9–11]. Its introduction in the early 2000s changed the more qualitative description of forests into a precise and accurate 3D representation with evident improvements in their quantification and monitoring [12], which, together with the fusion with data from other sources, are now creating new opportunities for upscaling the monitoring of the ecosystem structure [13]. More recently, UAV LiDAR has also been successfully used to estimate DBH [14,15], and its performances are comparable to those of ground-level TLS surveys [16]. These state-of-the-art LiDAR techniques can provide timely and accurate high spatial resolution measurements of forest 3D structure, facilitating data-driven, effective, and well-informed forest management [5,17,18]. In addition, the ground and the aerial LS systems provide complementary point clouds that are now being advantageously combined to provide a more comprehensive description of each tree [19,20]. However, the current price of TLS equipment or a well-designed UAV LiDAR system is still very high; besides, they demand specific technical expertise and superior experience from the users.

UAV photogrammetry is a methodology that is recently gaining recognition for its low cost, high performance, and flexibility in a wide range of applications concerning forest management, inventory gathering, and biomass estimation [7,21–24]. In their review of forest aboveground biomass (AGB) mapping methods, Réjou-Méchain et al. [7] highlight the importance of high-quality calibration/validation data and observe that UAV-based stereophotogrammetry has the potential to estimate AGB with accuracy close to that of ground measurements. Iglhaut et al. [21] present a recent review of SfM applications to forestry, in which they identify opportunities to extract structural attributes from airborne SfM point clouds, both in area-based and individual tree approaches. Goldbergs et al. [24] concluded that although airborne LiDAR provides more accurate estimates, SfM is an efficient and effective low-cost alternative to detect individual trees, measure tree heights, and estimate AGB in tropical savannas. The impact of UAV photogrammetry is being potentiated by the use of the structure from motion (SfM) technique [25], which originated from the field of computer vision [26] but incorporates the principles of traditional stereoscopic photogrammetry in its workflow. The workflow steps are as follows [27]: (1) identification of features (or "keypoints") in images of diverse perspectives and scales through the use of scale-invariant feature transform (SIFT) object recognition algorithm [28]; (2) identification of correspondences between keypoints in multiple images, and filtering to remove

geometrically inconsistent matches; (3) bundle adjustment to simultaneously estimate the 3D geometry of a scene, the camera poses, and camera intrinsic parameters; and (4) the application of multiview stereo (MVS) algorithms that increase the density of the point cloud by several orders of magnitude. Although, strictly speaking, SfM refers only to step (3) above, we are for the sake of simplicity naming the full workflow as SfM technique.

Although much less costly than LiDAR mapping, UAV photogrammetry has an important disadvantage (in comparison with LiDAR), which is the incapability of penetration of dense vegetation. This makes it more difficult to sense the elements located close to the ground and beneath the canopy, since each needs to be visible across various images. However, even if LiDAR outperforms SfM, according to the completeness of the reconstructed 3D model [29], the latter remains a cost-effective alternative in less dense forested environments [23,24]. Low flying heights and high image overlaps captured at off-nadir are needed in areas with higher canopy cover, in order to detect obliquely some subcanopy information with SfM [29–31].

Several studies are available in the literature that evaluate how UAV survey design parameters—such as image overlap or ground sampling distances—or processing software used impact SfM point cloud and image reconstruction [2,29,32,33]. Likewise, many studies have addressed SfM capabilities to produce below-canopy point clouds, thus achieving full 3D forest models [30–35], with a few authors addressing the estimation of individual trunk diameters [4,36]. Fritz et al. [36] pioneered the use of UAV-based SfM point clouds to measure trunk diameter at breast height (DBH) of individual trees in leaf-off state and called for additional research in what concerns vegetation state, flight design, and computer processing procedures. More recently, Ye et al. [4] attempted the estimation of DBH during both leaf-on and leaf-off seasons; they report poor quality of the SfM point clouds, especially in the leaf-on case, which was circumvented by the combination of point cloud datasets produced during four separate flights with different camera angles. In this work, we focus on the use of SfM photogrammetry with UAVs flying above the canopy only. UAV photogrammetry with under canopy flight is an exciting new approach that takes advantage of UAVs' obstacle-avoidance capabilities and has been shown capable to estimate DBH with accuracies comparable to those of TLS [37,38]. However, even if this method uses the same instruments, its field logistics and surveying design are closer to TLS surveying than to our approach. In the following, unless stated otherwise, when referring to UAV photogrammetry, we mean above-canopy UAV photogrammetry.

To our knowledge, no published study provides a systematic evaluation of how survey design and computer processing choices influence the retrieval of DBH with photogrammetry. The previous studies show that the estimation of this important bellow-canopy 3D structural parameter presents considerable challenges to UAV-based SfM, particularly in the case of trees with leaves. This paper contributes to filling this research gap by investigating the impact of flight design, acquisition parameters, and computer processing workflows on SfM capabilities to produce below-canopy point clouds and, particularly, to estimate the DBH of individual trees. The methodology was developed on a small piece of land within an agricultural field in Spain, where the results show that SfM can measure the DBH of single trees with very good accuracy. The best settings configuration found in the former experiment was successfully tested on a forested area in Bulgaria, with a mixture of leaf-on and leaf-off trees, where the DBH estimations achieved an RMSE (%) of 12.44%.

## 2. Materials and Methods

The methodology developed in this study consisted of five main steps: (1) UAV-based image acquisition for photogrammetry production, (2) manual tape measurements of trees' DBH for ground truth, (3) image processing and building of 3D point clouds, (4) estimation of DBH in the 3D point clouds, and (5) accuracy assessment of the UAV photogrammetry estimations. This methodology was applied for two study sites, albeit with different goals: at the first study area (Malaga, Spain), a small group of olive trees was selected to test many differently designed flights, aiming to identify which settings (flying altitude and

camera tilt) would give us the most accurate DBH estimations; then, in a forested area from Bulgaria (second study site), the best settings resulting from the previous work were used to assess the methodology in a different forest, encompassing a greater number of trees.

*2.1. Study Area*

The first study area is an agricultural field located in Andalusian Spain, in the province of Málaga (36°43′53″ N, 04°08′40″ W), mainly composed of orchards and olive trees. It is characterized by gentle to moderate slopes, with altitude ranging from 58 to 75 m above sea level, within an area of 0.7 hectares. We chose a row of 3 olive trees (Figure 1) for the UAV surveys. Olives 1 (towards the south) and 3 (towards the north) are composed of relatively short trunks overlaid by broad dense crowns and little understory. They reach approximately 5 m in height. Olive 2, in the middle, distinguishes itself from its neighbors by a low bifurcated trunk, presenting a challenge for DBH estimations.

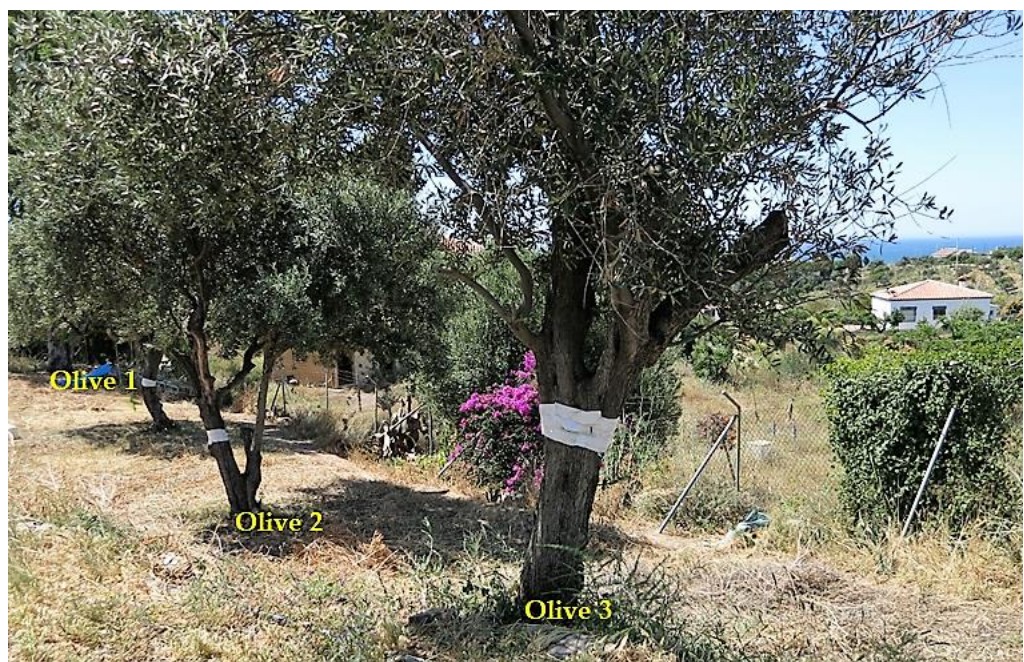

**Figure 1.** Olive trees surveyed in the first study area (Málaga, Spain).

The second study site is a forested area located in the foothills of Vitosha Mountain, on the outskirts of Sofia, the capital of Bulgaria. In this area, we tested the best settings configuration found at the first study site (Málaga). The surveyed site (Figure 2) is a 10 ha area adjacent to the southeastern border of Vitosha natural park, close to Zheleznitsa village, in Sofia province (42°32′44.15″ N, 23°21′46.79″ E). It is characterized by gentle terrain (altitudes from 1035 to 1050 m) and is covered by a grove of about 100 trees dominated by birch trees, a common species in the park. The trees are approximately 25 to 30 m tall and present relatively sparse canopy cover, since at the time of the surveys (early November) most trees had shed their leaves (Figure 3). The understory vegetation is small to moderate. DBH was manually measured and remotely estimated for 11 birch trees from this area.

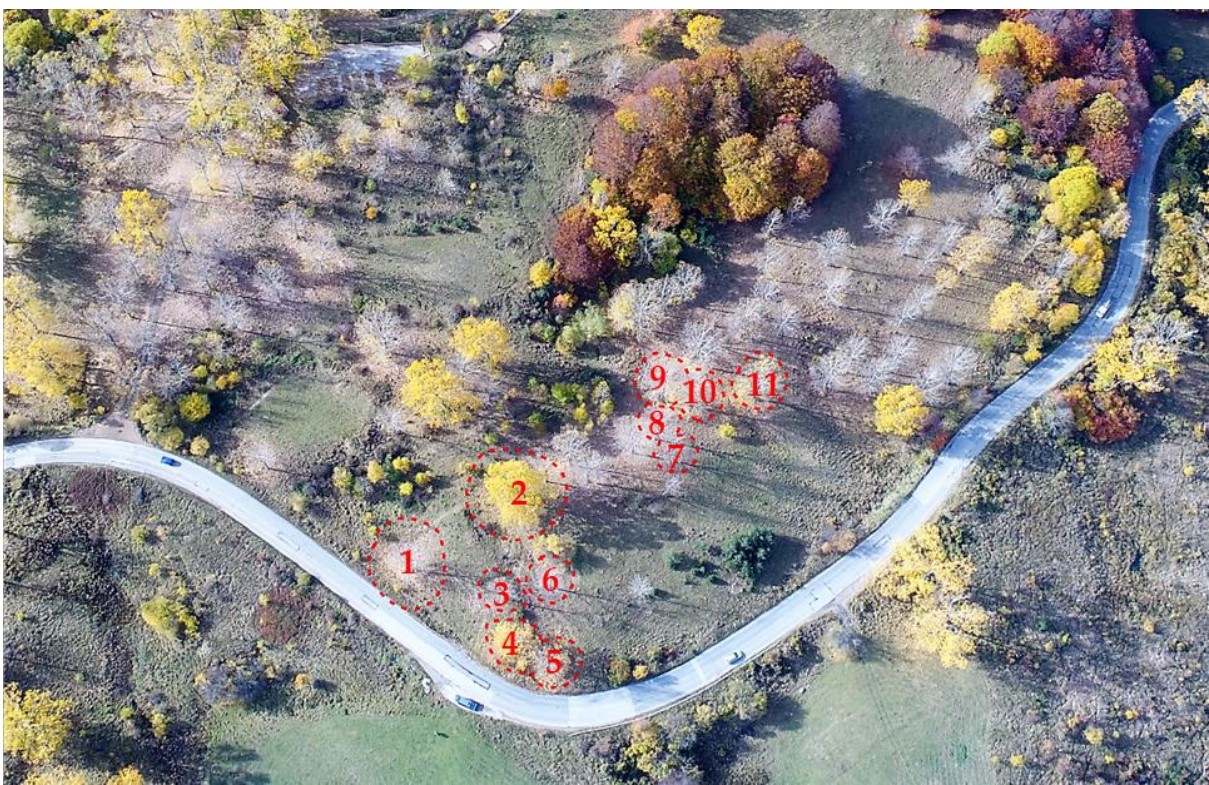

**Figure 2.** Aerial photograph of the second study site, an area adjacent to Vitosha natural park in Bulgaria dominated by birch trees. Eleven birch trees were surveyed for the DBH measurements and estimations.

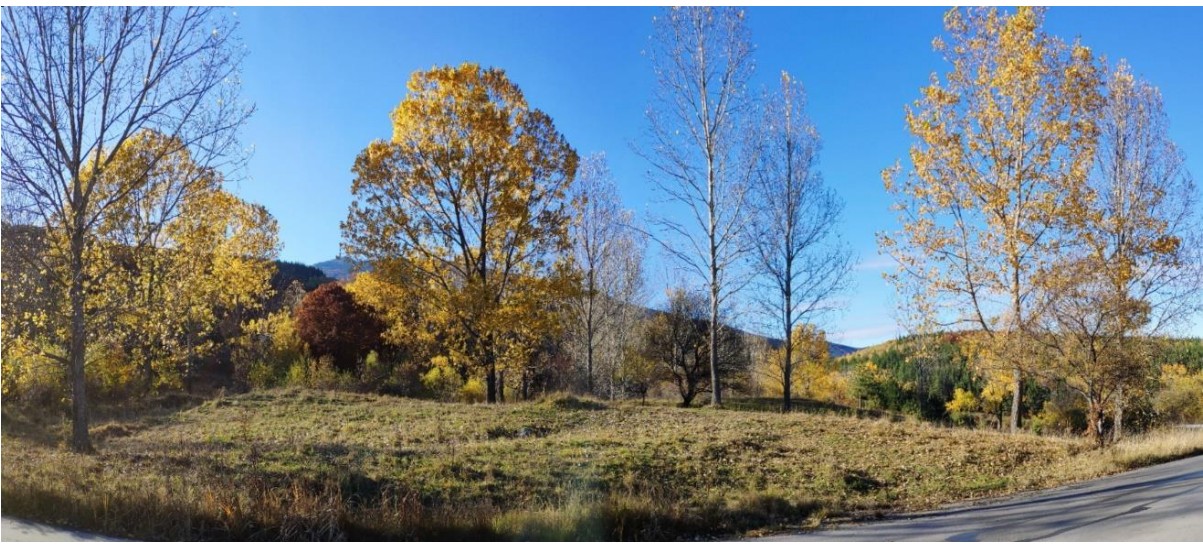

**Figure 3.** Representative picture of the second study site, an area adjacent to Vitosha natural park in Bulgaria dominated by birch trees.

*2.2. Equipment*

The image acquisition was performed using a Phantom 4 Pro UAV quadcopter, produced by DJI (Shenzhen, China). It weighs 1388 g, has a maximum flight time of approximately 30 min, and is equipped with an onboard satellite positioning system (GPS/GLONASS) and RGB camera. The Phantom 4 Pro camera is equipped with a 1-inch 20-megapixel CMOS sensor, mechanical and global shutter, and a 24 mm lens; it allows

adjustable aperture (f-2.8 to f-11) and has 5-directional obstacle sensing, an important advantage in forested areas.

### 2.3. UAV-Based Imagery Acquisition

The initial tests conducted at the Málaga study site indicated that when overlaps were chosen at values of 75–85%, point cloud reconstruction of the olive trunks was unsatisfactory for DBH estimation purposes, irrespective of changes in flight altitude or camera angle. For this reason, we fixed both the forward and side overlaps to 90%, while assessing the flying altitudes from 30 to 60 m above the terrain. The camera tilt was also varied from vertical (90°/nadir) to oblique (60°). The acquisition positions of the UAV camera followed double grid surveying lines (orthogonal lines), and the drone speed was set automatically according to the overlaps. These systematic experiments with Phantom 4 Pro were conducted throughout October 2020 between 10:00 and 14:00 (GMT+1) in order to analyze the images with the shadows at the same place. We used the application Map Pilot for iOS with the option "terrain awareness", using SRTM data as input. Table 1 lists the Phantom 4 Pro flight settings used at the Málaga site.

**Table 1.** Phantom 4 Pro flight settings assessed at the Málaga site.

| Flight Setting | Value | Type |
|---|---|---|
| Flight altitude (m) | 30; 40; 50; 60 | Variable |
| Camera tilt (°) | Vertical 90°; Oblique 60° | Variable |
| Forward overlap (%) | 90 | Fixed |
| Side overlap (%) | 90 | Fixed |
| Flight pattern | Double grid lines | Fixed |

In addition to the systematic flights conducted at Málaga using double grid surveying lines (Table 1), we also performed a low circular flight around each olive tree assessed (Table 2). The circular flight allowed detailed imaging of the tree structures while the drone flew around each of them, resulting in a very dense 3D point cloud model (Figure 4) that we assumed to represent, for benchmarking purposes, the best possible drone-derived photogrammetric reconstruction of the olive trees.

At the Vitosha mountain study site, the best settings configuration found at the former study site (Málaga) was tested. Here we placed markers on the ground before the flights, for which we collected their precise location with a D-GPS (EMLID RS2), obtaining in this way ground control points (GCPs). The DGPS was connected to the local NTRIP service, with the baseline located at less than 20 km, in order to improve the GCP position accuracies. The image acquisition took place at a flying altitude of 50 m (corresponding to approximately 25 m above the tree crowns), with forward and side overlaps of 90% and camera tilt angle of 60°. The 10 ha forested area was covered with double grid surveying lines during 4 flights lasting 30 min each at a flight speed of 2.1 m/s. In total, 1266 images were collected during a single day in early November (sunny, without clouds). The camera settings were set to manual, with shutter speed 1/500, ISO 400, and aperture adjusted to f3.5. Map Pilot (Maps Made Easy, San Diego, CA, USA) for iOS (with the option "terrain awareness") was used to plan and execute the flights. Table 3 lists the Phantom 4 Pro flight settings used at the Vitosha mountain site.

**Table 2.** Phantom 4 Pro circular flight setting used at Málaga site.

| Flight Setting | Value | Type |
|---|---|---|
| Flight altitude (m) | 20 | Fixed |
| Camera tilt (°) | Oblique 60 | Fixed |
| Flight pattern | Circular flight (manual mode) | Fixed |

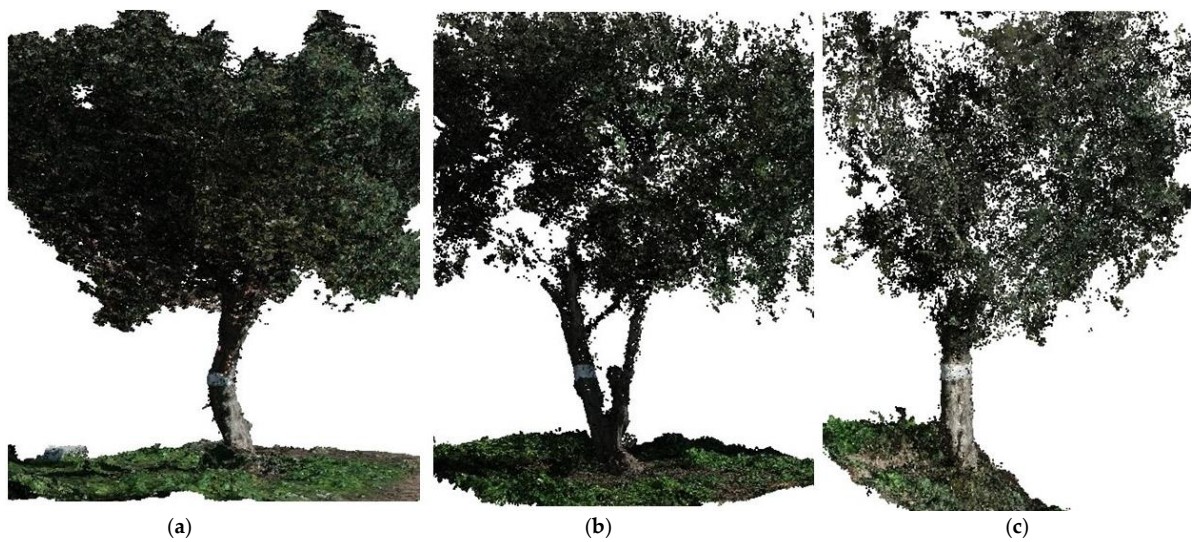

| (a) | (b) | (c) |

**Figure 4.** 3D point clouds of the olive trees surveyed in the first study area (Málaga - Spain). The figure shows the highly detailed point clouds obtained by the circular flights around each tree. The white stripes seen on the trunks were made as reference (height) for the DBH measurement. (**a**) Olive 1; (**b**) Olive 2; (**c**) Olive 3.

**Table 3.** Phantom 4 Pro flight settings used at Vitosha mountain site.

| Flight Setting | Value | Type |
|---|---|---|
| Flight altitude (m) | 50 (~25 m over tree canopies) | Fixed |
| Camera tilt (°) | Oblique 60 | Fixed |
| Forward overlap (%) | 90 | Fixed |
| Side overlap (%) | 90 | Fixed |
| Flight pattern | Double grid lines | Fixed |

*2.4. Field Data Acquisition (Ground Truth)*

All the trees analyzed in this study had their DBH manually quantified with a measuring tape to compare the ground truth values against the remote estimations, for accuracy assessment. The height typically used for DBH measurements is close to 1.3 m [39]. We had to adjust these values for some of the trees under investigation due to their nonconforming heights and shapes. At the Málaga site, the DBHs of Olives 1 and 2 were estimated at 1 m height, while Olive 3 DBH had to be measured at 0.6 m height due to its low and forked trunk. The birch trees in the foothills of Vitosha Mountain, on the other hand, had their DBH measured at about 1.4 m.

*2.5. Image Data Processing and Point Cloud Generation*

The images acquired by the UAV Phantom 4 Pro were processed using *Agisoft Metashape 1.6.4* and *Pix4D 4.6.4* software (Agisoft LLC, Russia, and Pix4D S.A., Switzerland). Both use similar SfM workflows (as defined in Section 1) to build 3D point clouds from a set of images. It is well known that, besides UAV survey design, which can for example determine the pixel size (ground sampling distance) of the images, computer processing methods (such as the software used or the workflow/parameters chosen) also impact the quality and density of the point clouds built.

To evaluate how different computer processing choices would influence our DBH estimation results, we chose two workflows for each software used, with differing levels of image resolution scaling and, hence, different densities of the point clouds generated (Table 4). With *Agisoft*, the point clouds were built using two different options: "*high*" (named here as AG-H), for which the resolution of the source image was downscaled with a ratio of 1:2, and "*ultrahigh*" (AG-UH), which keeps the source image at original size. With *PiX4D*, two similar configurations to those used in *Agisoft* were chosen to be assessed:

following the latter terminology, we designate them by *"high"* (PX-H), when the source image is downscaled 1:2 in PiX4D, and by "ultrahigh" (PX-UH) if PiX4D uses the original image size.

**Table 4.** SFM workflows used for image processing and point cloud building.

| Software | Processing Workflow |
|---|---|
| *AGISOFT* | AG-H (image scaling 1:2; point cloud density high) <br> AG-UH (image scaling 1:1; point cloud density ultrahigh) |
| *PiX4D* | PX-H (image scaling 1:2; point cloud density high) <br> PX-UH (image scaling 1:1; point cloud density ultrahigh) |

In Agisoft, the parameter for photo alignment was chosen at "high", while for building the dense cloud, the depth filtering was set to "mild".

### 2.6. DBH Estimation Point Clouds

The DBH estimations were carried out using the open-source *TreeLS R* package, developed specifically for forest monitoring applications and tree feature analysis from point cloud data [40,41]. The analysis with TreeLS was divided into 4 main steps: (1) filtering out ground points, (2) point cloud slicing at the desirable height (DBH), (3) trunk-point classification, and (4) circle fitting for DBH estimation.

The elimination of the ground points in the first step was performed using the *tlsNormalize* tool from the TreeLS package. Then, each individual tree within the point cloud was sliced 10 cm above and below the height corresponding to the ground truth DBH measurements (namely 0.6, 1.0, and 1.4 m, as explained in Section 2.4). To classify the points that represent the tree trunk, *TreeLS* uses a modified *Hough transform* method for circle search [40,41], a process that removes nonrelevant points such as small branches, understory vegetation, and noise (Figure 5b). The tool allows the entry of a maximum DBH to help the circle (trunk) search, for which we chose an overestimated DBH of 40 cm for Málaga and 80 cm for Bulgaria, considering the tree species of each study area.

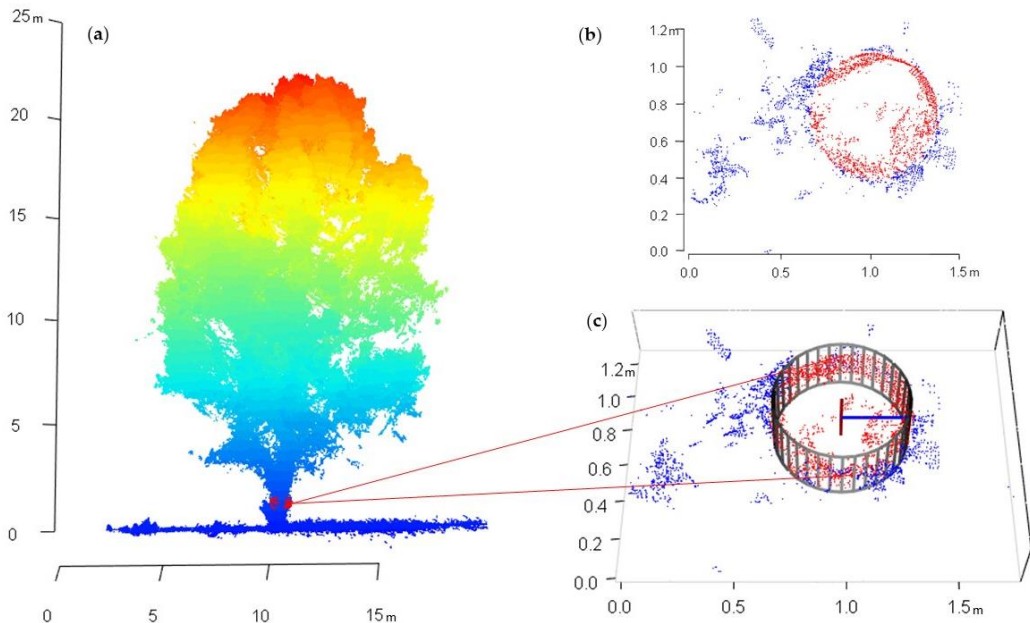

**Figure 5.** DBH estimation using 3D tree point cloud. (**a**) Perspective view of the whole tree colored by height, with trunk points already classified at the DBH (red trunk points). (**b**) Top view of the DBH slice already classified for trunk points (trunk points in red and nondesirable points in blue). (**c**) 3D view of the cylinder fitting tool applied for DBH estimations, using the previous classified DBH slice. This example represents a birch tree from our second study site, in Bulgaria.

To estimate the DBH in the sliced and classified trunk points, we used a circle fitting tool from *TreeLS* (Figure 5c). This tool applies a *least-squares circle fit* algorithm based on the *random sample consensus (RANSAC)* method to fit a circle (cylinder or sphere) to the classified trunk points [40,41]. To include the maximum number of trunk-classified points into the fitting cylinder, the inlier ratio was set to its maximum value (0.99).

### 2.7. Accuracy Assessment

To evaluate the accuracy of the DBH estimations, they were systematically compared against the manual measurements recorded in the field. The assessment was performed by calculating the absolute error, relative error (%), root mean square error (RMSE), and relative root mean square error (RMSE%) for each DBH value estimated, using the following equations:

$$\text{Absolute error} = |\hat{y}_i - y_i| \tag{1}$$

$$\text{Relative error (\%)} = \frac{|\hat{y}_i - y_i|}{\hat{y}_i} * 100 \tag{2}$$

$$\text{RMSE} = \sqrt{\frac{\sum_{i=1}^{n}(\hat{y}_i - y_i)^2}{n}} \tag{3}$$

$$\text{RMSE(\%)} = \frac{RMSE}{\bar{y}} * 100 \tag{4}$$

where $y_i$ is the measured value, $\hat{y}_i$ is the estimated value, $n$ is the total number of samples, and $\bar{y}$ is the mean of $n$ measured values [42,43].

## 3. Results

The following results allowed us to identify advantageous flight settings and processing methods for DBH estimations by UAV-based photogrammetry.

Some settings assessed (e.g., camera tilt of 90°) showed themselves to be not favorable for DBH estimations, often generating too poor (too few points) and, in other cases, too noisy point clouds. In the worst cases, the trunk classifier tool was not able to reliably find the trunk points, and so the DBH estimation could not be performed. Such cases can be identified at the bar chart results presented in the following with a "not available" (NA) symbol.

We firstly show the results from Málaga, analyzing each olive tree individually, followed by the RMSE considering all of them. Next are displayed the results from our second study area, in Bulgaria, where the best configuration found in Málaga was assessed for a distinct environment, with different tree species, encompassing a greater number of trees. The DBH estimations conducted in Bulgaria provided an RMSE (%) of 12.44%.

### 3.1. DBH Estimations—Málaga Study Site

First, all the photos in each of the different flights were used by Agisoft and Pix4D, in both high and ultrahigh modes, to build the point clouds. This means that the number of tie points between adjacent images is always high and, if needed, would result in orthomosaics and DSM where all the photos would be aligned. The ground sample distance (GSD) decreases with the increase of the altitude of the flights, as expected (Figure 6): this is more evident with Pix4D where the variations for high (0.88–1.63 cm) and ultrahigh modes (1.06–2.32 cm) are higher than with Agisoft, which is more restrained in both high (0.89–1.57 cm) and ultrahigh processing modes (0.89–1.74 cm).

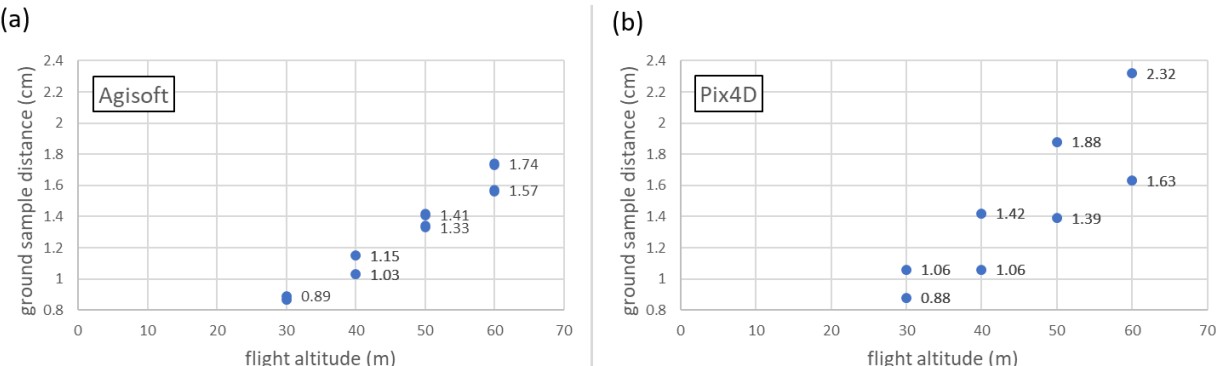

**Figure 6.** Variation of the ground sample distance with the altitude of the flights for the processing with (**a**) Agisoft and (**b**) Pix4D. In each pair of altitudes in both figures, the lower resolution value corresponds to high processing mode, while the other dot refers to the ultrahigh processing mode.

An interesting relative measure of comparison between the different models can be provided by the 2D density of points of the dense cloud (Figure 7). Although there is not a clear global pattern between the software and the processing modes, there are interesting results that can be derived from this feature. The density of points decreases in general with the altitude of the flight, a trend clearer for Agisoft than for Pix4D. Moreover, the density of points in each comparable pair is always higher for Agisoft than for Pix4D, except in the high processing mode for the nadir acquisition mode where the opposite is verified.

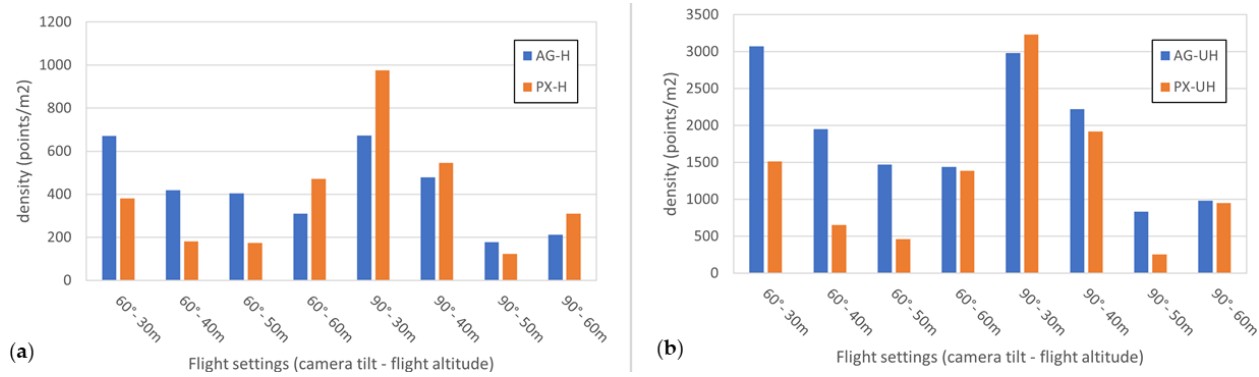

**Figure 7.** 2D point density of the dense clouds built after Agisoft and Pix4D procedures in (**a**) high processing mode and (**b**) ultrahigh processing mode.

The next chart (Figure 8) presents the relative and absolute errors for Olive 1 DBH estimations, showing that UAV flights using camera tilt of 60° degrees produced better results if compared to the tilt of 90°. The trunk of this low tree is situated right below the center of the canopy, hiding the trunk from most of the pictures captured by the UAV when using a camera tilt of 90°. Probably due to the same reason, lower flight altitudes could capture better pictures of the trunk, providing the best results with low flight altitudes of 30 and 40 m.

As regards computer processing methods, PX-UH produced better results in general for Olive 1, followed by AG-UH (Figure 8). Nevertheless, "60°–30 m" and "60°–40 m" UAV flight settings provided good results for all computer processing methods assessed, with estimation errors below 7.5%. This indicates that, when the flight acquisition design allows a good enough imaging of the tree trunks, computer processing choices do not play the dominant role in the quality of the results. That is shown clearly in the diagram in the case of the circular flight, with the lowest flying altitude and optimal image acquisition geometry.

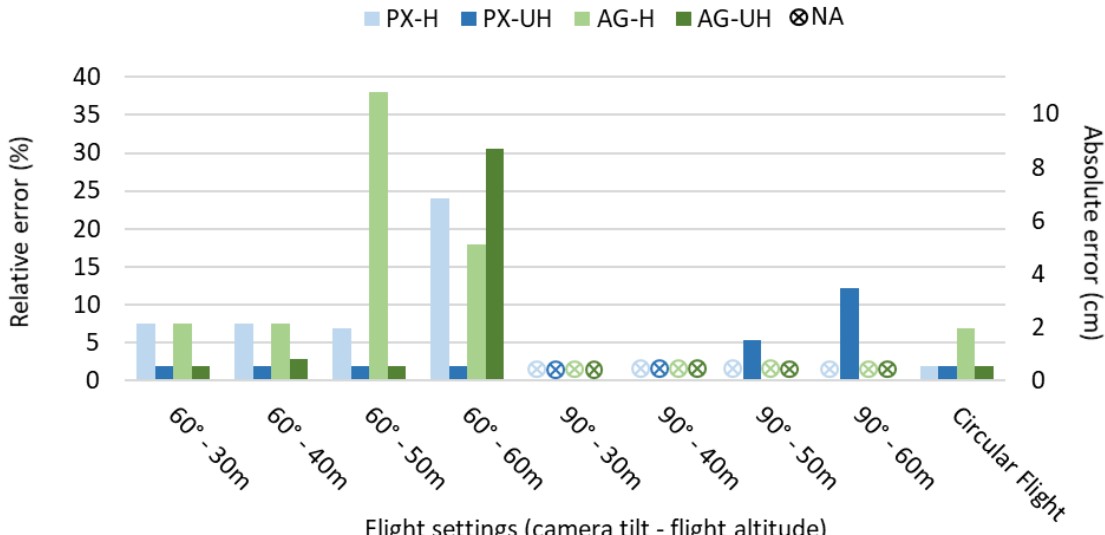

**Figure 8.** Bar chart with relative and absolute errors of the DBH estimations performed for Olive 1 from Málaga (DBH measured as ground truth: 28.5 cm). Each bar or NA symbol represents one point cloud used for the DBH estimation. Four processing methods with nine flight settings created 36 point-clouds for each tree from Málaga. The not available (NA) symbol represents too poor or too noisy point clouds in which the DBH estimations could not be performed.

Contrasting with Olive 1 (and Olive 3, as we will see in the following), Olive 2 presented better results with a camera tilt of 90° than with 60° (Figure 9). The reason is most probably the fact that its trunk is not centered below the canopy, but offset, and the result is that images captured with a tilt of 90° experience fewer difficulties with canopy occlusion. This could also explain why low altitude flights were not advantageous for imaging Olive 2, but instead, 60 and 50 m heights produced better results in this case.

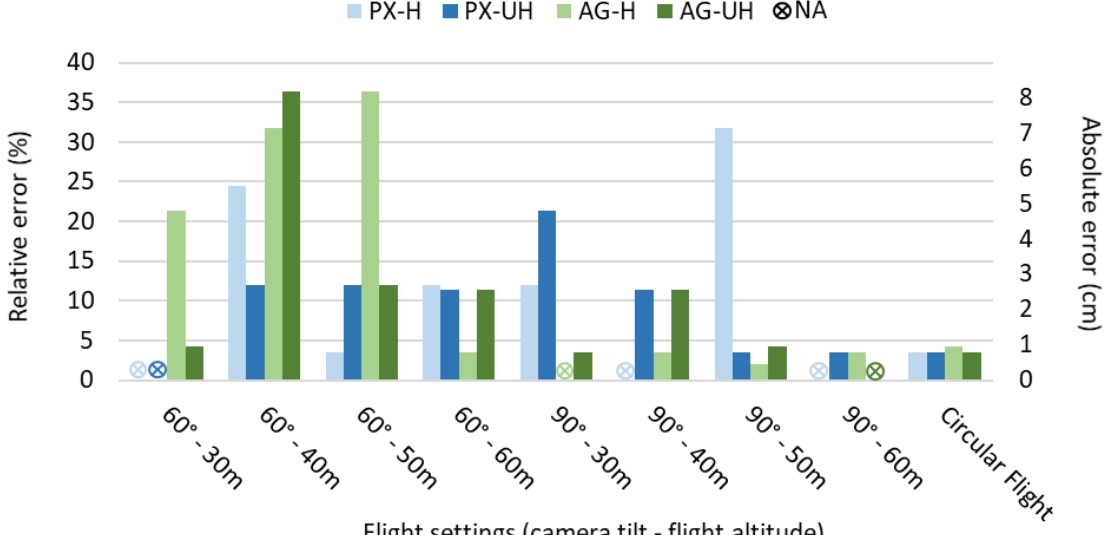

**Figure 9.** Bar chart with relative and absolute errors of the DBH estimations performed for Olive 2 from Málaga (DBH measured as ground truth: 22.4 cm). Each bar or NA symbol represents a point cloud used for the DBH estimation. Four processing methods with nine flight settings created 36 point-clouds for each tree from Málaga. The not available (NA) symbol represents too poor or too noisy point clouds in which the DBH estimations could not be performed.

In what concerns software workflow, PX-UH and AG-UH again produced better results, each displaying errors of over 20% only twice (Figure 9). The AG-H workflow, however, also presented good results, indicating that the lower point cloud density was in some cases an advantage for this tree by helping to alleviate the excessive noise frequently generated between the two main branches in the point cloud. The forked trunk of Olive 2 presented a very difficult shape for 3D modeling with UAV photogrammetry, resulting in noisy artifacts that the trunk-point classifier was frequently not capable of handling.

As was seen for Olive 1, Olive 3 presented the best results for DBH estimation with a camera tilt of 60° and lower flight altitudes of 30 and 40 m (Figure 10). For the point cloud building workflow, AG-UH stood out, presenting very good results for almost all flight settings.

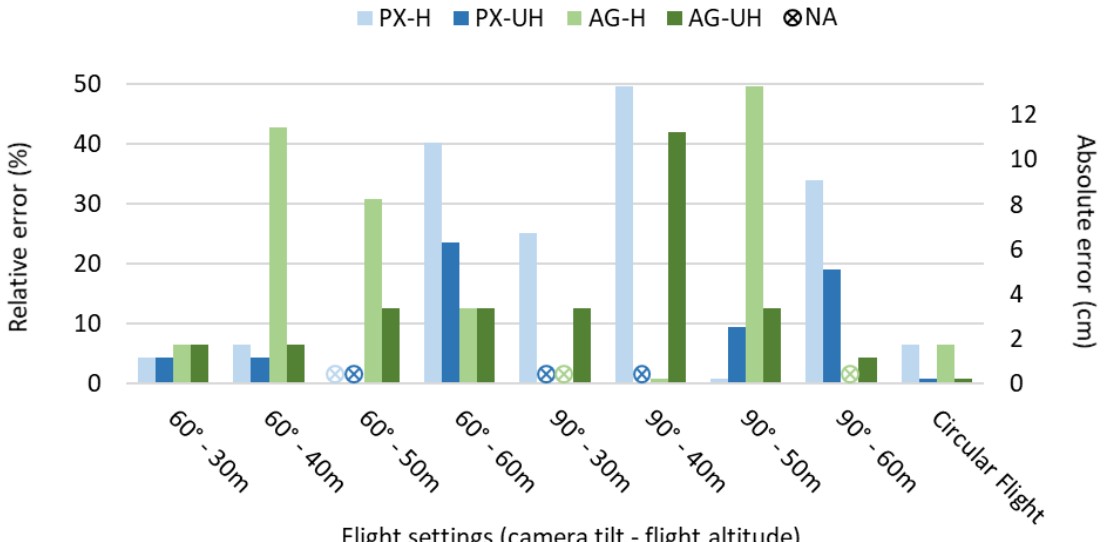

**Figure 10.** Bar chart with relative and absolute errors of the DBH estimations performed for Olive 3 from Málaga (DBH measured as ground truth: 26.7 cm). Each bar or NA symbol represents one point cloud used for the DBH estimation. Four processing methods with nine flight settings created 36 point-clouds for each tree from Málaga. The not available (NA) symbol represents too poor or too noisy point clouds in which the DBH estimations could not be performed.

To find the best settings configuration for DBH estimation using UAV-based photogrammetry, we calculated the root mean square error (RMSE) considering the three olive trees surveyed in Málaga. For calculating the RMSE, settings with "NA" results in any of the trees assessed were directly classified as NA. The charts in Figure 11 shows that camera tilt of 60° provided better results in general for grid survey flights and that the lower flight altitude assessed (30 m) was found to be the best choice when associated with a tilt of 60°. The ultrahigh processing methods (AG-UH and PX-UH) produced the smallest errors found, and the combination "60°–30m–AG-UH" stood out as the best setting configuration for grid survey flights. Figure 12 shows examples of point clouds and their DBH estimations obtained from three different settings.

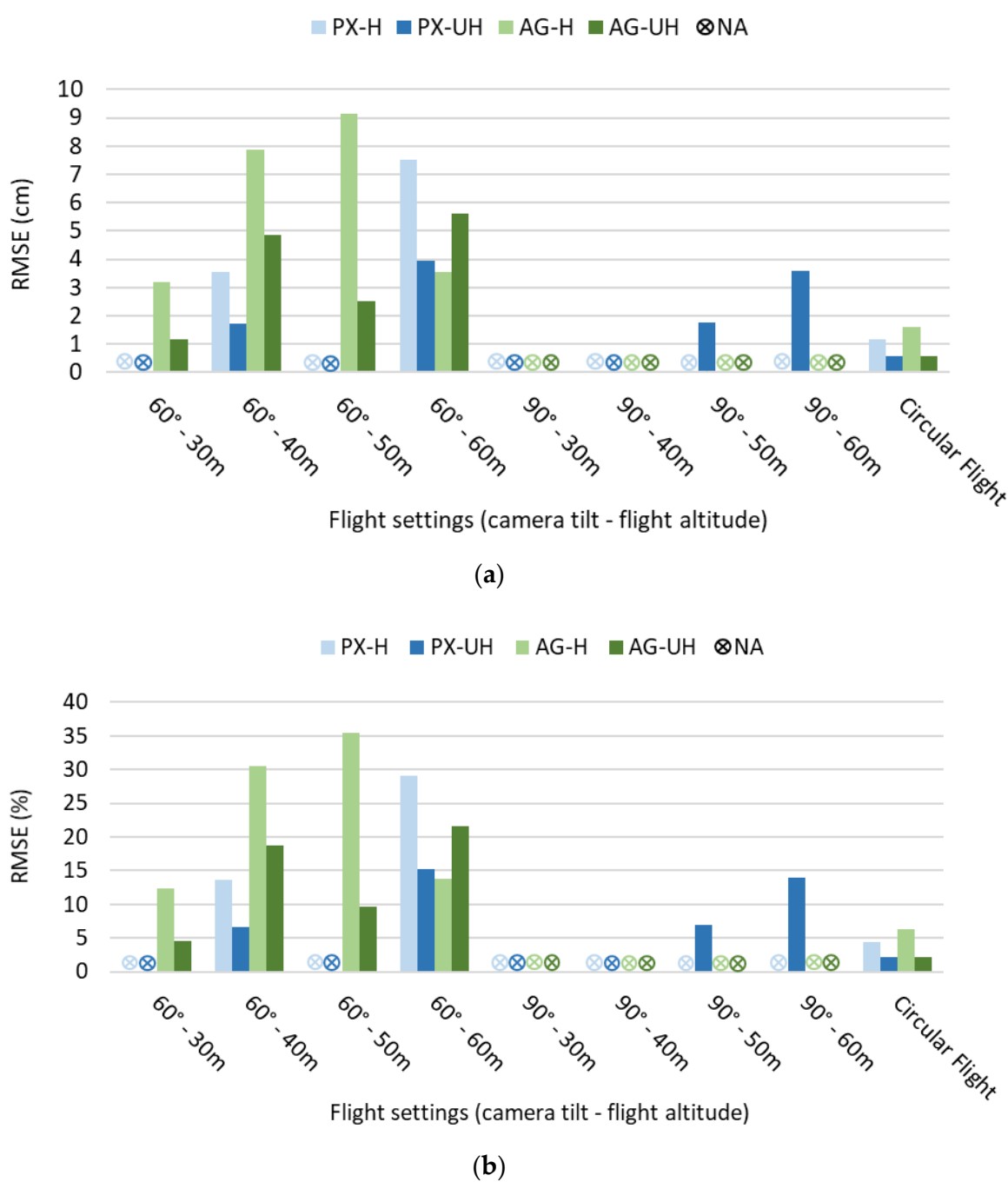

**Figure 11.** (**a**) Root mean square error (RMSE) and (**b**) relative root mean square error (RMSE%) of the DBH estimations carried out in Málaga. Four point-cloud building methods (PX-H, PX-UH, AG-H, and AG-UH), two camera tilts (60° and 90°), four flight altitudes (30, 40, 50, and 60 m), and circular flights were assessed.

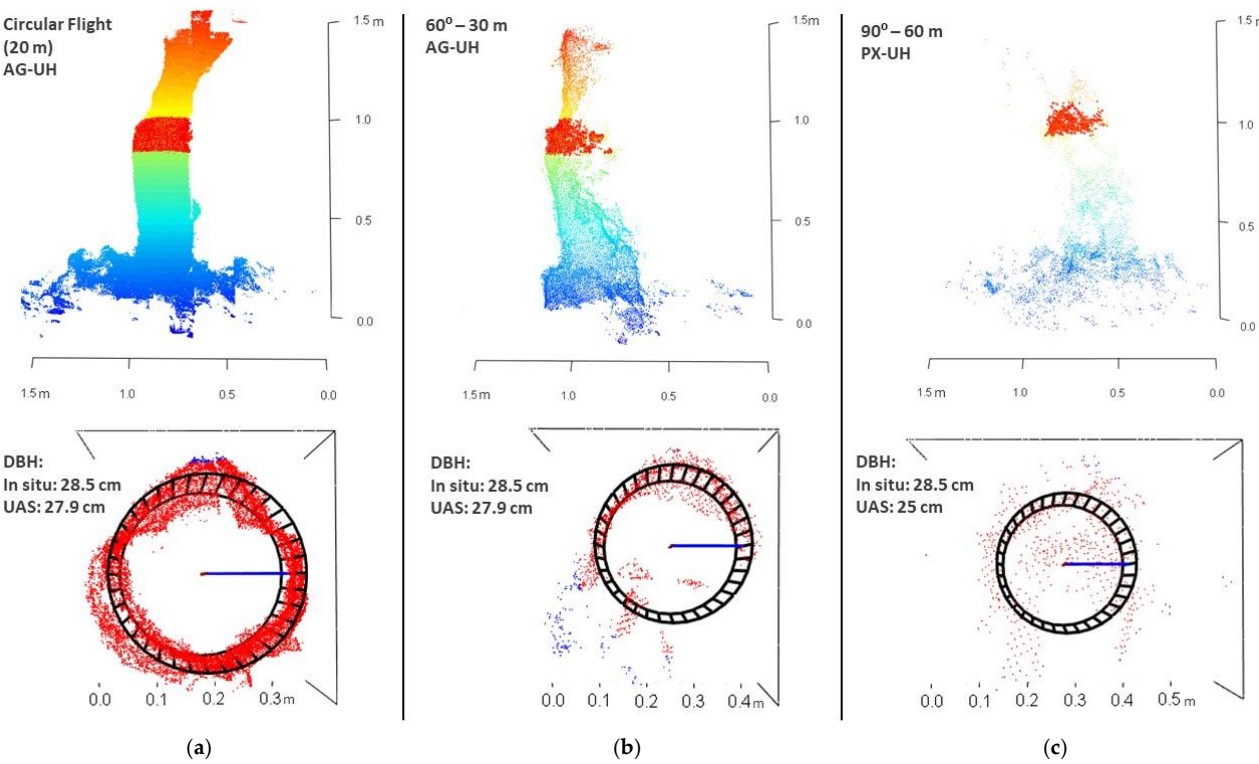

**Figure 12.** Examples of 3D point clouds of the trunk (perspective views above) and DBH estimations (cylinder top views bellow) of Olive 1, obtained by three different combinations of settings at Málaga: (**a**) circular flight (20 m) using AG-UH for point cloud building; (**b**) double grid flight survey using 60° (camera tilt), 30 m (flight altitude), and AG-UH for point cloud building; (**c**) double grid flight survey using 90° (camera tilt), 60 m (flight altitude), and PX-UH for point cloud building.

### 3.2. DBH Estimations—Bulgaria Study Site

The best settings configuration obtained in the previous study in Málaga (i.e., flight altitude 30 m, camera tilt 60°, and AG-UH processing method) was tested for a distinct forested area in Bulgaria, with other tree species (described in Section 2.1). The settings used were as following: flight altitude of 50 m (approximately 25 m above the tree crowns, similar to flying altitude of 30 m in the case of the 5 m tall olive trees), camera tilt of 60°, and AG-UH processing workflow. We obtained point clouds with and without using GCPs in their construction, for assessing their quality. Although GCPs provide more accurate products [44], the differences were negligible, so we decided to use only the models without GCPs in our evaluations, like in the Málaga study, where no GCPs were collected since the area was relatively small. Figure 13 exemplifies the point cloud obtained. Figure 14 allows a direct comparison between the DBH values acquired in the field and the ones obtained by remote estimations. As seen in Figure 15, all the DBH estimations of trees with available ground truth (11 in total) presented relative percent errors below 20%, and the global RMSE% for the same group of trees was 12.4%. Figure 15c also shows the strong relationship ($R^2 > 0.9$) found between the tape-measured and the remotely estimated DBHs. For this group of birch trees from Bulgaria, we observed no relevant difference in the accuracy of DBH estimation between leaf-on and leaf-off trees. Figure 16 shows two examples of point clouds and their DBH estimations, where both a leaf-on and a leaf-off tree point cloud can be observed.

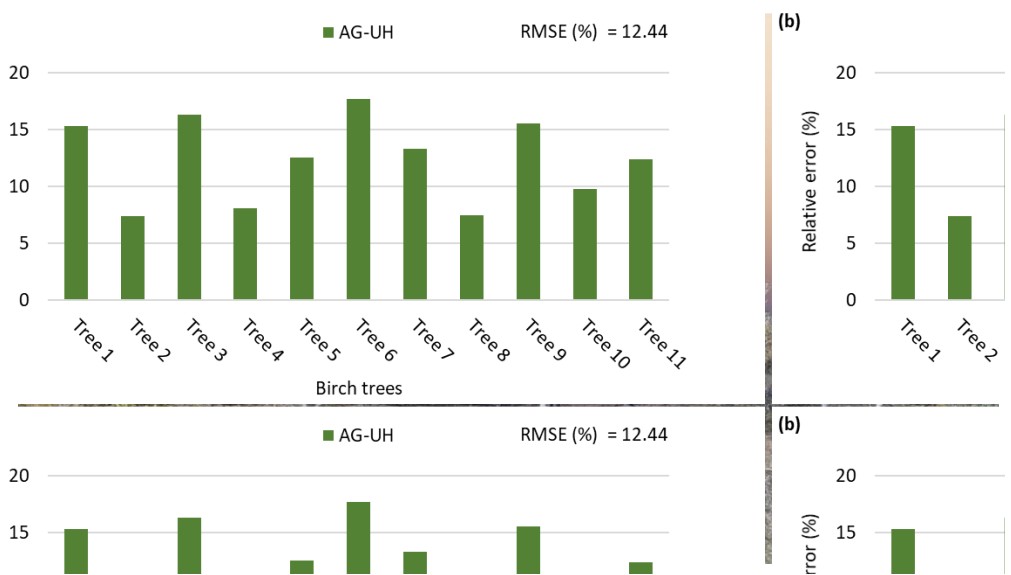

**Figure 13.** 3D point cloud of the second study site, an area adjacent to Vitosha natural park in Bulgaria dominated by birch trees.

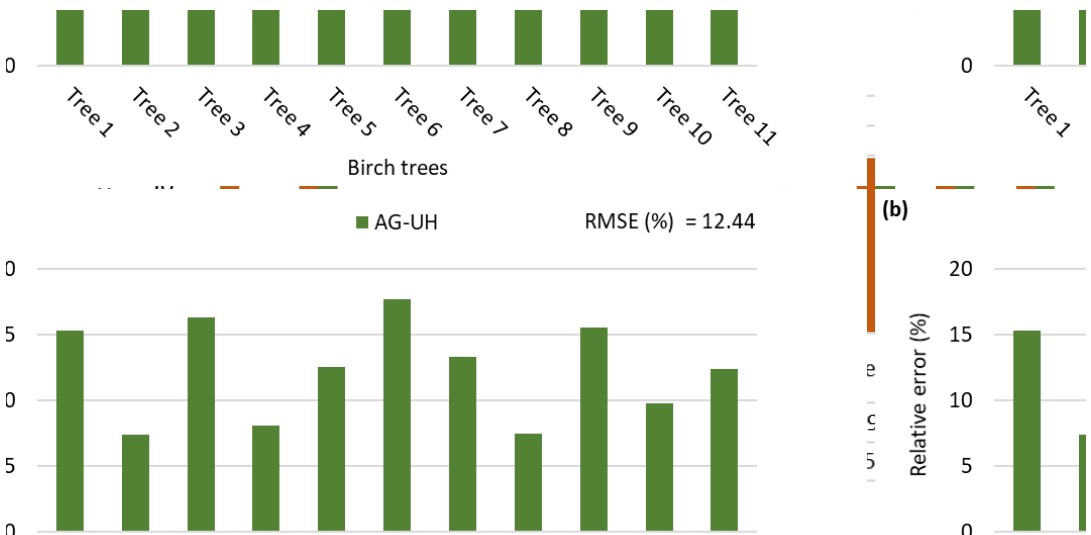

**Figure 14.** Tape-measured DBH values (in brown) and estimated values (in green) of 11 birch trees surveyed in the second study site (Bulgaria). All values are in centimeters.

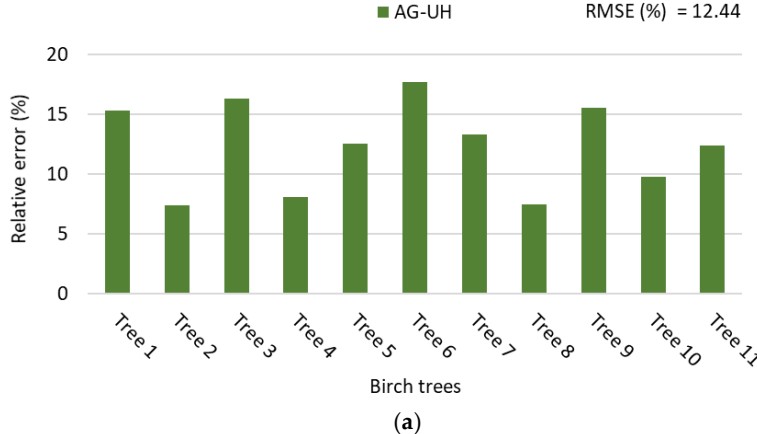

(**a**)

**Figure 15.** *Cont.*

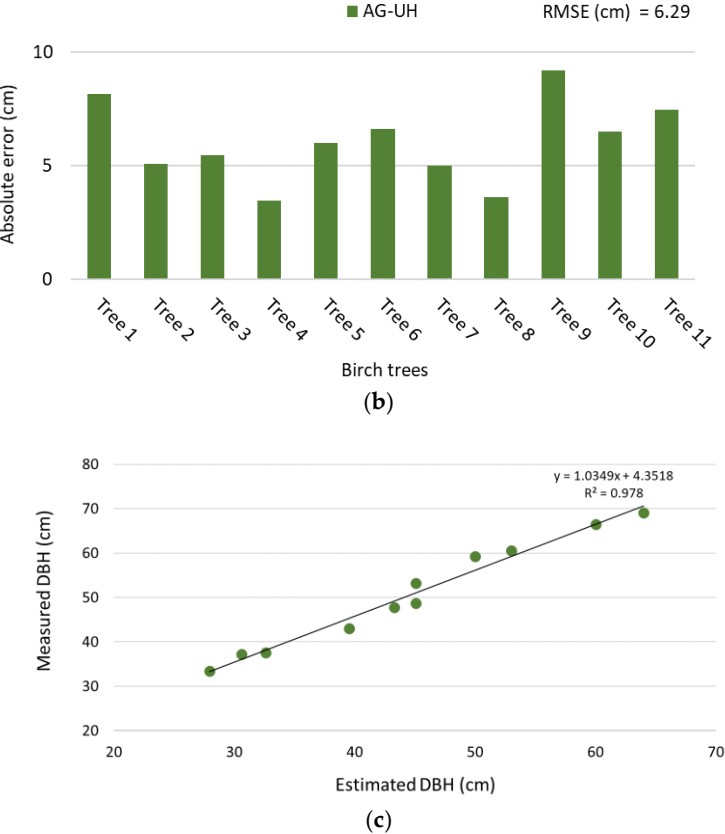

**Figure 15.** (**a**) Absolute errors (cm) and RMSE (cm) of the DBH estimations performed for the birch trees surveyed in Bulgaria. (**b**) Relative errors (%) and RMSE (%). (**c**) Scatterplot of the tape-measured DHB compared to the estimated DBH of the birch trees from Bulgaria. Eleven trees had their DBH recorded in the field with a measuring tape to be compared against the remote estimations. Trees 2 and 4 had most of their leaves, while the other ones were partially or totally without leaves.

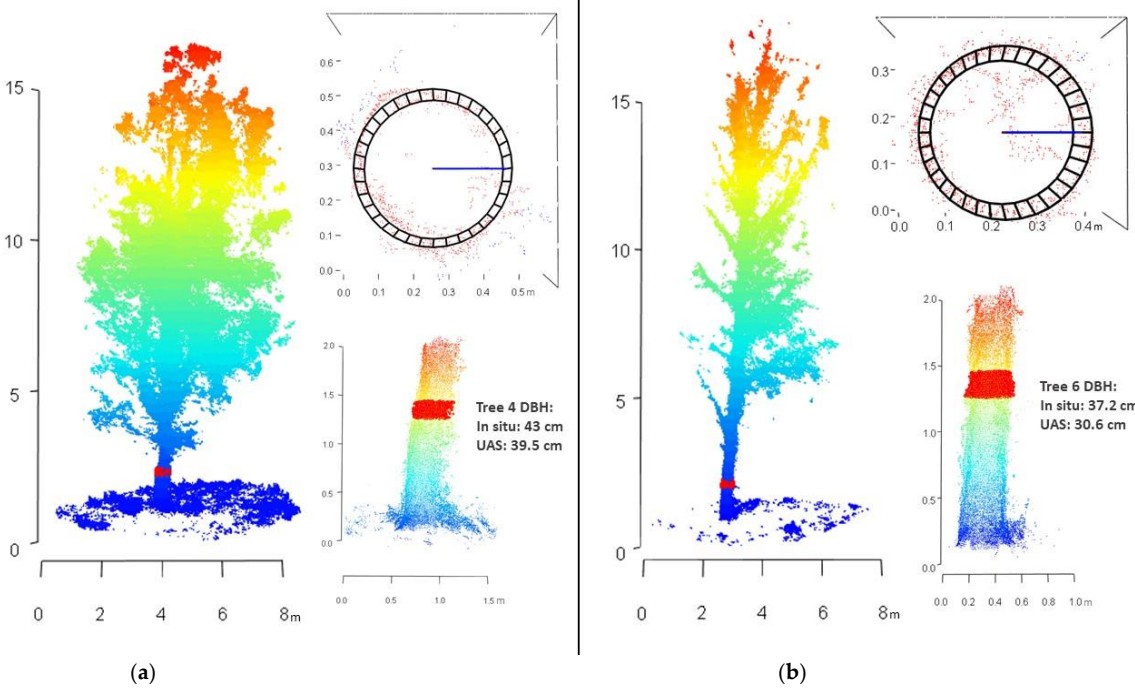

**Figure 16.** Examples of 3D point clouds and DBH estimations performed for the birch trees from Bulgaria: (**a**) Tree 2 (leaf-on); (**b**) Tree 6 (leaf-off).



## 4. Discussion and Conclusions

In this work, we investigated how flight design (namely flying altitude, camera tilt, and image overlap) and computer processing workflows impact the ability of UAV-based SfM photogrammetry to estimate DBH. We demonstrate that, when carefully designed methodologies are used, SfM can measure the DBH of single trees with very good accuracy; to our knowledge, the results presented here are the best achieved so far using (above-canopy) UAV-based photogrammetry [4,36].

Our experiments showed that a camera tilt of 60° improved the DBH estimations when compared to nadir camera orientation (90°), a result that is consistent with previous studies [4,45] and can be explained by the tilted camera's better viewpoint of the trunks totally or partially hidden under the canopies. Initial tests conducted in our study also indicate that forward and side overlaps below 90% are insufficient for point cloud reconstruction of the tree trunks, as previous works have demonstrated [42,46,47].

The flight altitude of 30 m (corresponding to about 25 m above tree canopies) produced the best results when associated with camera tilt of 60°, showing RMSE below 3.5 cm (15%) for Agisoft high processing method and below 1.5 cm (5%) for Agisoft ultrahigh processing method in our first study area, Málaga (Figure 11). The results of DBH estimation in Bulgaria (Figure 15), with the same flight settings applied and Agisoft ultrahigh processing method, also showed consistently low absolute and relative errors, an RMSE of 6.3 cm (12.4%). For context, ground-based manual measurement errors are typically sub-centimeter, but deviations around 5 cm are considered in biomass estimation models [48]. Ground-based TLS measurements, on the other hand, can estimate DBH with RMSE ranging from 3 to 5 cm, depending on the surveying design [49]. The other flight altitudes assessed in Málaga (40, 50, and 60 m) also provided some good results, but with less consistency, showing bigger errors depending on the condition assessed. We believe that when dealing with taller trees and open woodland areas (less canopy cover effect), as in the Bulgarian site, higher flight altitudes (such as 60 and 70 m, corresponding to ~35 and ~45 m above tree canopies) could sometimes, but not consistently, achieve good imaging of the trunks and produce satisfactory results.

Both software packages assessed were able to provide good tree point cloud models; however, for the best survey conditions found (camera tilt of 60° and flight altitude of 30 m) Agisoft provided superior robustness in the DBH estimation when compared to Pix4D. While Agisoft consistently showed satisfactory or good DBH results for all three olive trees surveyed in Málaga at all flight altitudes, Pix4D failed to provide acceptable estimation error for some of the flight altitudes or trees (Figure 11).

Ultrahigh processing methods produced very dense and more detailed point clouds, often allowing better DBH estimations. However, when the trunks were very well registered by the UAV pictures, as seen for the circular flights for example, high and ultrahigh methods presented just a slight difference in the accuracy. Considering grid surveying flights for forest inventory, where some trees would probably be poorly registered by some pictures, ultrahigh methods can bring important improvements for the DBH estimations, despite the higher processing power and time demanded.

In summary, flight altitude of 30 m (about 25 m above tree canopies), camera tilt of 60°, and Agisoft ultrahigh processing method were the best settings configuration found during this work for DBH estimation with (above-canopy) UAV-based photogrammetry. This configuration or similar is possibly replicable for other forested areas and different tree species, as shown by the study case carried out in the Bulgarian forest. More studies considering such different scenarios and a larger number of tree samples could increase the robustness of the data analysis. Nevertheless, some preliminary testing should be performed to fine-tune the set of parameters for each situation (depending on the type of trees and their spatial organization for example). However, the results presented here already provide important insights on how to obtain the best DBH estimations using photogrammetric procedures.

It is well known that the occurrence of shadows and uneven illumination can also affect the quality of the point cloud (e.g., [33]). Our study was not oriented towards a systematic assessment of the influence of light conditions, and future studies can better assess this issue by comparing sunny against cloudy days or comparing different hours of the day. We will test approaches that minimize these issues, namely the 3D reconstruction by the patch-based multiview stereo approach [50]. We will also test the most recent trend based on deep learning to deal with complex structures and smooth and less textured surfaces [51].

The main limitation of this method is the expected poor performance in areas with higher density of trees, as it depends on image registration of the trunk, which will be hidden by heavier canopy covers. In future work, we will assess how the DBH estimations obtained with this methodology are impacted by different tree densities, comparing with results obtained with UAV LiDAR systems. We expect our methodology to work well within low- to mid-density areas, such as urban and natural parks, forest meadows, or agroforestry systems, while only below-canopy UAV-based photogrammetry (e.g., see [37,38]) can compete with TLS or LiDAR and successfully estimate tree DBH for dense forest.

**Author Contributions:** Conceptualization, B.M.M., G.G., P.P., and S.H.; methodology, B.M.M., G.G., P.P., and S.H.; software, B.M.M., G.G., and S.H.; validation, B.M.M., G.G., P.P., O.V., and S.H.; formal analysis, B.M.M., G.G., P.P., and S.H.; investigation, B.M.M., G.G., P.P., and S.H.; resources, G.G., P.P., O.V., and S.H.; data curation, B.M.M., G.G., and S.H.; writing—original draft preparation, B.M.M. and S.H.; writing—review and editing, B.M.M., G.G., P.P., O.V., and S.H.; supervision, P.P. and S.H.; project administration, S.H.; funding acquisition, P.P. and S.H. All authors have read and agreed to the published version of the manuscript.

**Funding:** This research was funded by FCT (Fundação para a Ciência e a Tecnologia—Portugal), grant number PTDC/EAM-REM/30475/2017.

**Institutional Review Board Statement:** Not applicable.

**Informed Consent Statement:** Not applicable.

**Data Availability Statement:** Not applicable.

**Conflicts of Interest:** The authors declare no conflict of interest.

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
