# Peer review of "Assessment of the Influence of Survey Design and Processing Choices on the Accuracy of Tree Diameter at Breast Height (DBH) Measurements Using UAV-Based Photogrammetry"

_drones, doi:10.3390/drones5020043_

Round 1
Reviewer 1 Report
The authors examine the impact of flight planning and software on the generation of 3D point clouds in order to measure DBH in two monospecific tree stands. Overall the research question is interesting, but there are some methodological flaws that need to be addressed. As well, the authors need a more clear and thorough description of SfM, there are some errors throughout the intro that must be fixed/clarified. The two main methodological concerns that impact the results and make the direct comparison difficult are 1) no GCPs used to improve the point clouds. This is important because the UAVs used here are equipped with only a basic GPS so from that perspective the point cloud will have errors not only in terms of its positional accuracy but also the size of objects will be incorrect and this could have a large impact on their results. 2) the varying illumination conditions also are a concern. In treed areas especially the position of shadows has been shown in the literature to impact the overall outcome of SfM generated point clouds.
Specific comments
Intro
Line 79: That is incorrect, see original references e.g. S. Ullman The interpretation of structure from motion Proc. R. Soc. Lond. Ser. B, 203 (1979), pp. 405-426
Lines 82-86: This paragraph is misleading, while there are similarities there are definite differences between SfM and conventional photogrammetry, see:
Smith, M.W.; Carrivick, J.L.; Quincey, D.J. Structure from motion photogrammetry in physical geography. Prog. Phys. Geogr. 2016, 40, 247–275
Overall, the authors need a more thorough and accurate description of SfM in the intro or methods. Simply listing 5 studies in line 81, grouped together in a reference isn’t enough. What did these studies find? What is the main outcome?
Line 85: Too simplistic, authors need to introduce the SIFT algorithm, keypoints, and it’s not just pose information that is retrieved and optimized, there are internal camera characteristics as well
Line 88: Several studies have shown the opposite is true. LiDAR is a mature technology with minimal room for error by the operator as opposed to SfM where naïve mistakes during the photograph acquisition or processing lead to suboptimal point clouds and results.
Overall authors need to be more careful with their terminology, SfM refers to only the first stage of the generation of tie-points. One must them apply a densification algorithm such as MultiView Stereo to achieve the final results. Earlier on in the intro authors need to be clear what they mean and not use terms such as SfM incorrectly. If they want to use SfM for brevity, then define earlier on what they mean and how they will use it going forward.
Line 108: computer processing choices is vague
Figure 1, albeit an important figure, is a result, it should not be in the methods. Replace with an actual photograph.
Figure 3 – Same comment as above for Figure 1 – move to results.
Section 2.3 – is a difference between the two cameras also rolling vs global shutter?
Line 181 – ‘at routine values’, what does that mean? 75%-85% is very low for vegetated areas.
Normally the flight speed is set based on the altitude, overall and acquisition parameter such as equal distance or equal time. Flight speed is not normally a variable that is adjusted for this purpose because the camera is triggered automatically. Also what is meant by grid pattern? For vegetation it has long been accepted by that orthogonal lines achieve better detail and more accurate point clouds. Table 1 seems to contradict the paragraph above regarding the settings.
Line 190: How did the varying shadows affect the results?
Line 196: mention of double-grid here contradicts las paragraph on page 5
Table 3: Why only oblique?
Were any GCPs used in the point cloud generation? Neither of the UAVs used have a very accurate GPS which results in both positional errors of the point cloud but also within model errors in terms of the dimension of objects and their distances. See Kalacska M, Lucanus O, Arroyo-Mora JP, Laliberté É, Elmer K, Leblanc G, Groves A. Accuracy of 3D Landscape Reconstruction without Ground Control Points Using Different UAS Platforms. Drones. 2020; 4(2):13.
Line 228: Incorrect, while both follow a similar workflow their implementation of SIFT and other steps varies greatly.
Results.
Lines 287-302: Statements needs to be backed up by numerical results.
Figures 5, 6, 7: Error bars?
More rigorous quantitative analysis/statistics should be reported when comparing the various combinations. In addition, authors should report basic point clouds stats such as number of points, number of matched keypoints, etc which would allow the reader to make a more informed decision about the quality of the point clouds.
Lines 442-451: The issue here comes back to the methods where authors state they mixed various illumination conditions.
The reference list overall is quite thin for such a large set of complex questions.
Author Response
Dear reviewer 1, please see the attachment.

Reviewer 2 Report
Dear authors, my remarks and suggestions are in the attachment. Good luck.

Author Response
Dear reviewer 2, please see the attachment.

Reviewer 3 Report
The main issue with the experiment is a very low number of trees used within the experiment. Based on three olive trees is chosen the data acquisition protocol. This is too small to make any suggestions. Then in the next experiment 11 trees is also a very small amount. Also, establishing protocol within three olive trees and then use it in different conditions does not make sense to me. I believe it could show that different flight parameters could be better or with the same accuracy on the second site.
Another issue that is maybe more important is the inconsistency of the experiment. For olive trees multiple flights with a straight line and then one flight at 20m circular. So it is not possible to compare the flight patterns. And then authors chose 50m above with grid lines.
I do not see a point to mention the Phantom 3 adv. When you have not used it to create data. It can be moved to discussion if you want to address it. But also you have not used the same parameters as with Phantom 4 so to compare results with Phantom 3 the experiment must be the same for both.
I would say the idea is good but it must be plan more carefully and consistently. Also, the number of trees to make assumptions must be increased to be able to make some more general conclusions. In this stage, I do not see a taka a home message for readers due to the inconsistency and the very small amount of trees.
I would update the idea and also work on additional products from UAV photogrammetry flights – orthomosaic, DTM, DSM. To bring something more than the terrestrial photogrammetry is providing. But at the same time, you need to be honest about where it can be used. It is impossible to use it in dense forest. So maybe within urban trees or something similarly sparse.
L30-31: Please read a Kuželka et al. (2018) where they achieved similar results within a dense forest. But they have used under canopy flight. The statement “the best achieved so far using UAV-based photogrammetry“ is not correct in my opinion. Maybe UAV-based photogrammetry above the canopy.
Papers to add within Introduction and Discussion:
Kuželka, K.; Surový, P. Mapping Forest Structure Using UAS inside Flight Capabilities. Sensors 2018, 18, 2245. https://doi.org/10.3390/s18072245
Krisanski, S.; Taskhiri, M.S.; Turner, P. Enhancing Methods for Under-Canopy Unmanned Aircraft System Based Photogrammetry in Complex Forests for Tree Diameter Measurement. Remote Sens. 2020, 12, 1652. https://doi.org/10.3390/rs12101652
Also, you should consider papers dealing with terrestrial photogrammetry.
Author Response
Dear reviewer 3, please see the attachment.

Round 2
Reviewer 1 Report
The authors have clarified and improved their overall manuscript. However, a few questions remain.
Line 64: need a reference for the 10pt/m2, in my experience that is very high from manned aircraft, normally we see 1-4pts/m2
Line 120: No matter the flight height, an RBG photograph will not ''see" through the canopy
In general, the use of the term 'drones' is colloquial, best to pick something more technical, UAV, UAS, RPAS, etc. (e.g. line 135)
Line 138, generally we can't 'prove' something simply because it is written in another paper, statements can be 'shown' however - this is an important distinction
Line 213, put the time zone (e.g., GMT + 2) rather than calling it Spain time
Figure 4 is out of place, it is a result, and should be moved to the results section.
Line 236, did the RS2 have any incoming corrections (e.g. NTRIP) to improve the position of the points? If so, how far was the baseline? If not, was it operated in base station mode with one unit acting as the base (preferably left onsite for several hours to obtain a more precise position) and a second unit as a rover? Was PPP used? Were the positions post processed to a base station - were the results a FIX solution? Without any corrections, the basic positional error of these units is quite large, on the order of meters.
Line 335: first sentence not needed
Line 340: do you mean ground sampling distance?
Line 444: what does 'the differences were residual' mean? Is potentially the issue that the GCPs themselves have relatively high positional error?
Line 450: Figure 15c shows an R2, not a correlation
Figure 15 c: 4 decimals not needed, R2 alone is not enough, add the actual function
Lines 526-529: unclear
